# Liver Transplantation for Incidental Cholangiocarcinoma or Combined Hepatocellular Carcinoma/Cholangiocarcinoma—Own Experiences and Review of the Literature

**DOI:** 10.3390/cancers15143609

**Published:** 2023-07-13

**Authors:** Laura Schwenk, Oliver Rohland, Aladdin Ali-Deeb, Felix Dondorf, Utz Settmacher, Falk Rauchfuß

**Affiliations:** Department of General, Visceral and Vascular Surgery, Jena University Hospital, Am Klinikum 1, 07740 Jena, Germany; oliver.rohland@med.uni-jena.de (O.R.); aladdin.ali-deeb@med.uni-jena.de (A.A.-D.); felix.dondorf@med.uni-jena.de (F.D.); utz.settmacher@med.uni-jena.de (U.S.); falk.rauchfuss@med.uni-jena.de (F.R.)

**Keywords:** liver transplantation, intrahepatic cholangiocarcinoma, combined hepatocellular cholangiocarcinoma, transplant outcome

## Abstract

**Simple Summary:**

The diagnosis of intrahepatic cholangiocarcinoma in a cirrhotic liver is considered a contradiction for transplantation in Germany, as well as many other international transplantation programs. The aim of our retrospective study was to evaluate the long-term outcomes of patients with incidental combined hepatocellular- and cholangiocarcinoma and sole intrahepatic cholangiocarcinomas after liver transplantation. Between January 2010 and December 2022, iCCA was found in eight patients post-transplant. We confirmed high overall survival and low recurrence rates after liver transplantation. It can be stated that liver transplantation in the case of combined hepatocellular carcinoma and sole intrahepatic cholangiocarcinoma presents a possible curative therapy option.

**Abstract:**

Background: Data about liver transplantation for mixed tumors from hepatocellular carcinoma to cholangiocarcinoma are limited. Furthermore, the diagnosis of intrahepatic cholangiocarcinoma or combined tumors in a cirrhotic liver is considered a contraindication for transplantation. Our aim was to evaluate the long-term outcomes of patients with incidental cholangiocarcinoma or combined tumors after liver transplantation. Methods: In our descriptive analysis, data were evaluated from all patients since 2010 who received a liver transplant due to an assumed hepatocellular carcinoma at Jena University Hospital. Survival rates were determined using the Kaplan–Meier method. Results: Between January 2010 and December 2022, an incidental intrahepatic cholangiocarcinoma was found in eight patients post-transplant. Four combined hepatocellular and cholangiocarcinoma and four sole intrahepatic cholangiocarcinomas were found. A recurrence through distant metastases from combined hepatocellular- and cholangiocarcinoma was found in one patient at one year after transplantation. Another patient developed a pulmonary primary tumor independently one year post-transplant. The recurrence rate was at 14.3%. While two patients died, the 1- and 5-year overall survival rates post-transplant were 87.5% and 75%, respectively. Conclusion: Patients with intrahepatic cholangiocarcinoma or combined hepatocellular- and cholangiocarcinoma could profit from liver transplantation.

## 1. Introduction

Intrahepatic cholangiocarcinoma (iCCA) is a malignant entity originating from the epithelial cells of the intrahepatic bile ducts. Defined by its anatomic localization (proximal to the right and left hepatic duct), iCCAs are differentiated from perihilar cholangiocarcinoma, distal cholangiocarcinoma and gallbladder carcinoma within the current guidelines for the diagnosis and therapy of HCC and biliary carcinoma [1]. In total, iCCA accounts for 10–15% of all primary liver carcinoma [2]. In some cases, iCCA is still misdiagnosed due its similar radiological features with HCC [3]. The distinction between HCC and its malignant differential diagnosis is important. When a biliary differentiation component is present alongside hepatocellular differentiation, the diagnosis of a combined hepatocellular cholangiocarcinoma (cHCC-CCA) is given [1].

cHCC-CCA is a rare liver malignancy and accounts for 2–5% of all primary liver carcinoma [4]. This tumor is known to have both hepatocytic and cholangiocytic differentiation within the same lesion [5,6]. Compared with classic hepatocellular carcinoma (HCC), this biophenotypic malignancy has a more aggressive course and a poorer prognosis [7]. 

Since the first report of cHCC-CCA by Wells in 1903, the incidence increased continuously over the decades, as well as the clinical importance of the tumor [8]. 

Nevertheless, its similarity to HCC makes the preoperative diagnosis of cHCC-CCA or iCCA challenging. iCCAs or cHCC-CCAs are often coincidentally found in the final postoperative pathological analysis, such as on patients who received a liver transplant due to a PSC or misdiagnosed HCC in cirrhosis [9]. Data about effective treatment and predictable occurrences are currently difficult to obtain due to the absence of clear terminology in the literature. Currently, resection with lymph node dissection is the only curative option for patients with cHCC-CCA [10]. While liver transplantation offers a globally curative therapy option for non-metastatic HCC, data about liver transplantation for mixed tumors from HCC and cholangiocarcinoma (cHCC-CCA) are limited. Furthermore, the diagnosis of iCCA in a cirrhotic remodeled liver is considered a contradiction for transplantation in Germany as well as many other international transplantation programs [11]. According to the current guidelines, liver transplantation for iCCA should not take place outside of studies [1]. This is due to the aggressive behavior of this tumor entity, with early tumor recurrences and low survival rates [1]. Nevertheless, studies repeatedly show the advantage of liver transplantation in the case of iCCA and cHCC-CCA [12,13,14]. The aim of this study is to evaluate the long-term results of patients with cHCC-CCA after liver transplantation.

## 2. Materials and Methods

Data were evaluated from all patients at Jena University Hospital between 2010 and 2022 who received a liver transplantation due to an assumed hepatocellular carcinoma. In eight cases, the postoperative histopathological findings showed a combined hepatocellular- and cholangiocarcinoma or sole intrahepatic cholangiocarcinoma.

The following data were analyzed: overall survival (OS); disease-free survival (DFS); recurrence rate; and need for re-transplantation. Patient diagnoses, neoadjuvant therapy, type of transplantation, donor characteristics, tumor information such as size and differentiation, pre- and postoperative tumor markers and general patient data in the context of clinical, surgical and pathological findings were collected from the university hospital database SAP (SAP Global Corporate Affairs, Walldorf, Germany). Selected data of patients are presented as medians and ranges. Survival studies were determined using the Kaplan–Meier method. Overall survival was defined as the duration between liver transplantation and patient death. For all analyses, SPSS statistics (IBM Corp., Armonk, NY, USA) and Microsoft Excel (Microsoft Corporation, Redmond, WA, USA) were used.

To access the current main studies regarding liver transplantation for cHCC-iCCA, we performed literature research and discussed this in the context of our results using the following search keywords: “liver transplantation”, “hepatocellular carcinoma”, “intrahepatic cholangiocarcinoma”, “neoadjuvant therapy”, “combined hepatocellular cholangiocarcinoma”. The electronic databases included: PubMed, Google Scholar and MEDLINE. 

## 3. Results

Between January 2010 and December 2022, 686 deceased donor liver transplantations and 145 living-donor liver transplantations were performed. A total of 191 patients received a liver transplant due to an assumed hepatocellular carcinoma. Among the 191 patients, iCCA was found in eight patients post-transplant. Of these patients, four cHCC-CCA and four sole iCCA were found. One patient received a transplant through living-donor liver transplantation. Post-mortem-donor characteristics are shown in Table 1, as well as living-donor characteristics (Table 2).

Patients’ median age at the time of transplantation was 60.25 years (range = 29–74 years). All patients were male. Four patients were within the Milan criteria and eight exhibited signs of liver cirrhosis. The tumor diameter ranged between 1.3 and 6 cm (Table 3). Post-operative tumor classifications are shown in Table 3. The median pre-operative serum levels of alpha-fetoprotein, carcinoembryonic antigen and carbohydrate antigen 19-9 were 171 ng/mL (range = 2–842), 2.63 ng/mL (range = 1–5) and 57.25 U/mL (range = 1–188), respectively (Table 3). The preoperative, 3-month and 12-month median levels of carbohydrate antigen 19-9 are shown in Figure 1. 

Through the assumption of HCC, three patients were bridged with transarterial chemoembolization (TACE). One further patient was bridged with TACE and radiofrequency ablation. One patient received selective internal radiotherapy, while another was treated with stereotactic radiotherapy. In two cases, no bridging therapy was administered. 

The mean operation time was 300 min, while the intraoperative blood loss was between 600 and 1500 mL. The median intensive care duration was 4.5 days, and the median length of hospital stay was 27 days. All bile duct anastomoses were reconstructed using duct-to-duct anastomosis. For one patient, within one year post-transplant, conversion into a biliodigestive anastomosis was necessary. Within 30 days post-operation, one patient showed a bile leakage, which was treated through an endoscopic stent insertion. Another patient required dialysis due to acute kidney failure. Furthermore, a respiratory insufficiency was diagnosed in one patient, based on acute respiratory distress syndrome, which was treated with reintubation and, subsequently, percutaneous tracheostomy. 

In total, two patients needed a re-operation. The reason was a thrombosis of the hepatic artery and also an abdominal compartment syndrome, followed by transplant failure. The 30-day high-grade complication rate (Clavien ≥ IIIa) was 62.5% (Table 3). Within 90 days post-transplantation, two patients received a drainage due to a pleural effusion and ascites. The 90-day high-grade complication rate (Clavien ≥ III) was 28.5% (Table 3). Of eight patients, one showed a mild transplant rejection, which was successfully treated through five-day cortisone therapy.

The median follow-up amounted to 36 months. A relapse through distant metastases (pulmonary, pleural) from combined hepatocellular- and cholangiocarcinoma was found in one patient at one year after transplantation. It was treated with chemotherapy (four courses of Gemcitabine and Cisplatin) as well as radiotherapy. Even two years after recurrence, the patient is still alive. Another patient developed a pulmonary primary tumor independently one year post-transplant. The 1-, 3- and 5-year recurrence rates were 14.3%. The 1-, 3- and 5-year overall survival rates after liver transplantation for patients with cHCC-CCA and iCCA were 87.5%, 75% and 75%, respectively (Figure 2 and Figure 3). Two patients died during the observation period. The causes of death were primary graft dysfunction two days after liver transplantation and the occurrence of a primary tumor within the lung one year post-transplant. Due to a failure of the liver transplant, one patient received a re-transplantation two days after the initial operation. The 1-, 3- and 5-year disease-free survival rates were 75%, 62.5% and 62.5%, respectively.

## 4. Discussion

Even though liver transplantation in the case of HCC is globally accepted as a curative therapy, it remains disputed in the context of cHCC-CCA, as well as iCCA, and poses a contraindication. This is mainly because of poor overall survival, high recurrence rates and the aggressive behavior of these tumor entities. This assertion was proven through various studies; an overview of the analyses between 1994 and 2015 and their results is shown in Table 4. 

Park et al. performed liver transplantation on 2137 patients between January 1999 and December 2009. In all cases, HCC was diagnosed pre-operatively. Post-operatively, the histologic examinations led to diagnoses of cHCC-CCA in 15 cases. In the evaluation of long-term outcomes, seven patients suffered from a tumor recurrence; six of these occurred within the first year post-transplantation. To conclude, liver transplantation as a therapy option in the case of cHCC-CCA was challenged by the authors [15]. Similarly, dissatisfactory results were presented by Hara et al. in a multi-center study. Within the study, the results of 6627 liver transplantations from 45 institutions were analyzed between January 2001 and December 2015. In 12 transplantation centers, 19 cases of incidental iCCAs were reported, which were treated through liver transplantation. In 10 of 19 patients, a relapse occurred post-transplantation. Due to these results, the authors also concluded that iCCAs are associated with a high recurrence risk and poor prognosis, even in the case of an incidental finding of the tumor within the liver [16]. Even though various studies show doubt over liver transplantations in the case of cHCC-CCA and iCCA, there are increasing long-term results in the literature which confirm liver transplantation to be an alternative therapy with good outcomes and similar recurrence rates for cHCC-CCA. 

After evaluating our results, we could not confirm the doubts about liver transplantation in the case of cHCC-CCA and iCCA. In our retrospective analysis, we showed a high overall survival and low recurrence rate after liver transplantation. Out of the eight transplant patients, four were diagnosed with cHCC-CCA and four with sole iCCA. Within the combined group, only one recurrence of previously known carcinoma occurred within the first year. In the group of sole iCCAs, no recurrence or lethality has occurred to this date. Observing the presented data, it can be said that liver transplantation due to cHCC-CCA and sole iCCA offers a possible curative therapy option for selected patients.

Furthermore, various publications present satisfactory transplantation results for cases of iCCA in the early stages [17], as well as developed stages under the condition that a neoadjuvant therapy was successful [18]. 

Itoh et al. also presented liver transplantation to be a curative therapy option for patients with cHCC-CCA [19]. However, the condition for this is the fulfillment of Kyushu University’s criteria or the Milan criteria. In their study, long-term results of living-donor liver transplantation for 178 patients between 1999 and 2014 were evaluated. Eight of these patients were diagnosed with cHCC-CCA. In all eight cases, the Kyushu University criteria were fulfilled, while the Milan criteria were fulfilled in six cases. In our study, we could not apply the Kyushu University criteria due to missing laboratory parameters. The survival and recurrence rate post-transplantation for patients with cHCC-CCA and HCC did not differ statistically. The overall 1-, 3-, 5- and 10-year survival rates and the disease-free survival after transplantation for patients with cHCC-CCA were nearly equivalent to patients with sole HCC [19]. Similar results were published by Facciuto et al. in 2015. They reported 32 patients with iCCA or cHCC-CCA that received a liver transplant [20]. Their results showed that patients diagnosed with cHCC-iCCA or iCCA which fulfilled the Milan criteria reached a similar prognosis compared to the respective HCC patients [20]. Especially concerning iCCA with a diameter of up to 2 cm, the results of liver transplantation for iCCA/cHCC-CCA are similar to those of liver transplantation in the case of HCC [21]. In addition to that, another analysis of the National Cancer Database showed the similarity between liver transplantation in the case of iCCA and liver resections. The Kaplan–Meier-Analysis resulted in a 5-year overall survival rate of 36.1% for patients who received a liver transplant, compared with 34.7% in the case of a liver resection [22]. De Martin et al. also evaluated the similarity between liver resection and liver transplantation due to cHCC-CCA and iCCA in a study in 2020 [23]. In this study, 75 iCCA/cHCC-CCA patients (all in cirrhosis) with a maximum tumor diameter of 5 cm were observed, of which 49 received a liver transplantation and 26 received a liver resection. An interesting point is that liver transplantation showed improved results compared with liver resection in regard to 1-, 3- and 5-year recurrence rates. The overall survival rate was nearly the same in both groups. Moreover, the study confirmed the similarity between transplantation results regarding overall survival and disease-free survival with tumors of up to 2 cm and larger tumors of 2–5 cm. Additionally, the authors showed through their multivariate analysis that the results of transplantation correlate more with tumor differentiation than tumor diameter [23]. 

Nevertheless, a patient selection through the Milan criteria is established for liver transplantation of HCC patients. In the case of cHCC-CCA and iCCA, criteria have not been determined to this date. The significant correlation between poor tumor differentiation and tumor recurrence was presented already [11,23]. Sapisochin et al., as well as Vallin et al., also postulated that vascular invasion directly impacts transplantation results [11,24]. Furthermore, Vallin et al. determined that tumor recurrence occurred in 100% of cases with vascular invasion, in comparison to 0% of cases without vascular invasion [24]. In the international retrospective study performed in 2016 by Sapisochin et al., it was also pointed out that microvascular invasion has a significant correlation with tumor recurrence, shown within both the univariate as well as the multivariate analysis [11]. 

Unfortunately, we could not show any correlation between recurrence rate and tumor differentiation or tumor diameter in our analysis. 

The response to neoadjuvant therapy poses another important factor for possible patient selection. This controls the development of the disease and offers an opportunity to undertake liver transplantation on previously inoperable patients. Antwi et al. examined the role of loco-regional therapy before transplantation in 19 cHCC-CCA patients. The results presented an improved 3-year overall survival rate in cHCC-CCA patients that responded to neoadjuvant therapy, in comparison to non-responders [25]. In summary, such characteristics could be of significance regarding the selection of patients with cHCC-CCA and iCCA for liver transplantation. 

In our study, four patients were bridged with TACE. Two patients showed a good response to treatment, while one patient showed no response and one further patient was transplanted before the success of the therapy could be evaluated. All patients who were bridged with TACE did not relapse regardless of their response to therapy. A relapse from combined hepatocellular- and cholangiocarcinoma was found in one patient at one year after transplantation. The patient was bridged with stereotactic radiotherapy. Response to bridging therapy could not be evaluated because transplantation followed one month after radiotherapy. Interestingly, the two patients who did not receive bridging therapy showed no signs of recurrence. In summary, we could not find any correlation between bridging therapy and survival. 

## 5. Conclusions

Observing the previously presented data, it can be said that liver transplantation in the case of cHCC-CCA and sole iCCA presents a possible curative therapy option for selected patients. In regard to the exhibited meaningful data from the literature, as well as based on our results, the conservative assumption of classifying patients with cHCC-CCA or iCCA as not transplantable due to poor survival rates and increased relapse rates should be dismissed. Much more significance is found in the selection of patients through appointed criteria for admission to liver transplantation in the case of cHCC-CCA and iCCA. However, to confirm this thesis and select appropriate criteria, further studies are necessary.

## Figures and Tables

**Figure 1 cancers-15-03609-f001:**
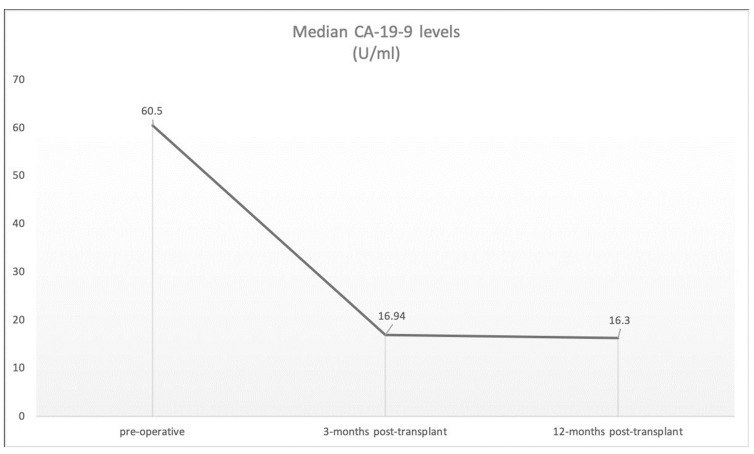
Median CA-19-9 levels.

**Figure 2 cancers-15-03609-f002:**
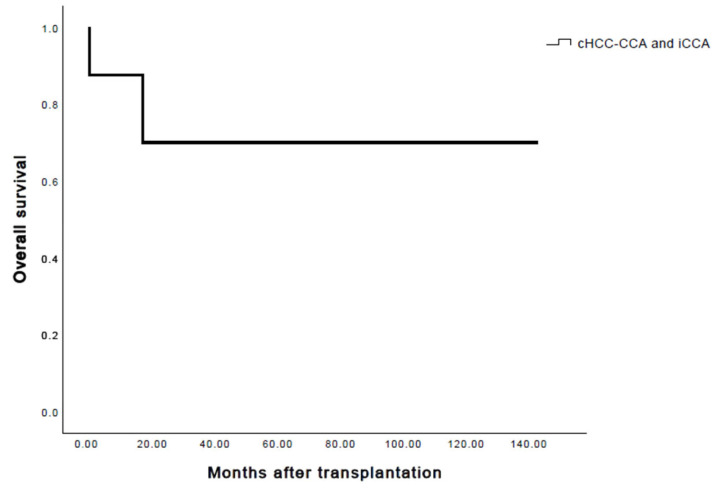
Combined overall survival, cHCC-CCA and iCCA.

**Figure 3 cancers-15-03609-f003:**
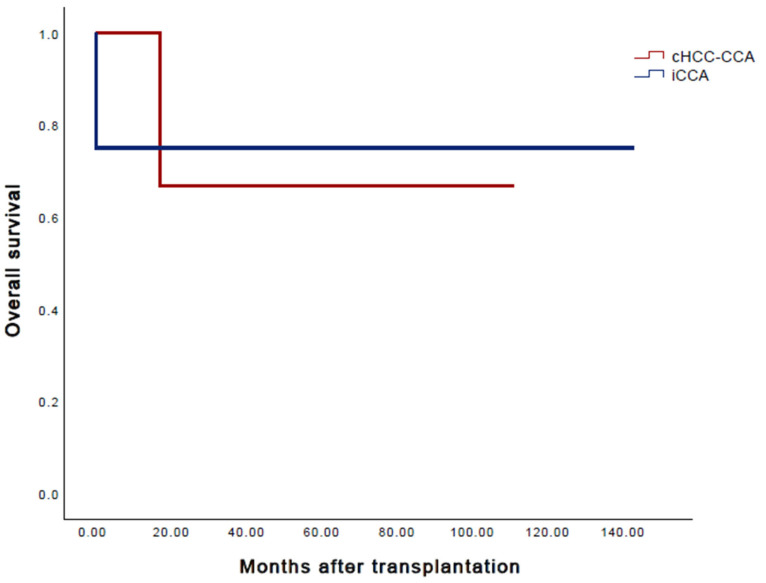
Overall survival, cHCC-CCA and iCCA.

**Table 1 cancers-15-03609-t001:** Post-mortem-donor characteristics.

	Age (Years)	Height (cm)	Weight (kg)	BMI (kg/m^2^)	ICU Stay (Days)	Steatosis %	ASAT/ALAT Pre-Explant µmol/L*s	Sodium mmol/L	Bilirubin µmol/L	Viral Hepatitis	Sepsis	Meningitis	SARS-CoV-2-Infection	Malignant Tumor
I	57	163	76	29	12	no	33/48	139	4.9	negative	negative	negative	n.a.	post-explant-lung
II	66	173	60	20	7	no	64/46	147	17.4	negative	negative	negative	n.a.	negative
III	48	190	90	25	3	yes (61–70% microvesicular steatosis)	102/84	131	3	HBs Ab positive	negative	negative	negative	negative
IV	62	180	85	26	3	no	109/25	128	13.7	negative	negative	negative	n.a.	negative
V	72	170	90	31	4	no	29/23	132	14	negative	negative	negative	n.a.	negative
VI	57	180	90	28	3	no	28/24	135	10.3	negative	negative	negative	n.a.	negative
VII	76	175	80	26	6	no	129/53	141	4	negative	negative	negative	negative	negative

**Table 2 cancers-15-03609-t002:** Living-donor characteristics.

	Age (Years)	Height (cm)	Weight (kg)	BMI (kg/m^2^)	Blood Type	Relation to Recipient	Right or Left Lobe of the Liver	Mass of the Graft (cm^3^)
I	51	164	68	25	0	mother	right	1009

**Table 3 cancers-15-03609-t003:** Characteristics of patients.

	Diagnosis	Localization	Child–Pugh SCORE	Blood Type	Meld Score	Largest Tumor Diameter (cm)	Number of Lesions	Tumor Classification	CA19-9 Pre- Transplant (U/mL)	CA19-9 3 and 12 Months Post- Transplant (U/mL)	AFP Pre- Transplant (ng/mL)	AFP 3 and 12 Months Post- Transplant (ng/mL)	CEA Pre- Transplant (ng/mL)	CEA 3 and 12 Months Post- Transplant (ng/mL)	Clavien–Dindo Complication Classification (30 days/90 days)	Rejection/Transplant Dysfunction	Dialysis Post-Transplant
I	cHCC-CCA	segment IV	A	0	10	6	1	not further classified	42.9	27.8/13.4	468.9	2.4/3.9	<1	0.9/0.9	IIIb/0	no	no
II	cHCC-CCA	segment II/III	B	A	12	3.5	1	pT1a, pN0 (0/4), L0, V0, Pn0, R0 G3	188	23.5/16.8	29.4	1.3/2.7	4.4	1.2/n.a.	IVa/IIIa	no	yes
III	cHCC-CCA	segment V/VI	A	A	12	3.3	1	ypT2, pN0 (0/2), L0, V0, Pn0, R0	24.4	22/21	842	3.2/2.9	2.1	0.9/1.5	II/0	mild rejection	no
IV	cHCC-CCA	segment VI, VII, VIII, IV, V	A	0	12	4	3	HCC: ypT2, pNx, L0, V0, Pn0, R0 iCCA: ypT1a, L0, V0, Pn0, R0	32.7	8.9/8.9	2.7	2.7/3.4	5	2.2/2.3	0/0	no	no
V	iCCA	segment V	B	AB	22	5.5	1	pT1, pN0 (0/3), PMx, L0, V0, Pn0, R0, G2	146	12.9/12.9	6.2	4.8/3.6	1.4	0.9/0.9	IIIb/0	no	no
VI	iCCA	segment VI	A	A	8	4	1	not further classified	25	-	3.8	-	0.9	-	V/-	transplant dysfunction	yes
VII	iCCA	segment IVb	B	AB	9	1.7	1	pT1a, pN0 (0/2), L0, V0, Pn0, R0, G1	12.7	12.8/8.9	9.7	3.9/2.7	4.3	2.4/2.5	IIIa/0	no	no
VIII	iCCA	multiple	B	A	19	1.3	multiple	ypT2, ypN0 (0/1), L0, V0, Pn0, R0, G2	12.3	10.7/32.2	11.6	5.3/2.8	2.5	3.1/3.6	0/IIIa	no	no

**Table 4 cancers-15-03609-t004:** Overview of previous analyses.

Serial Number	Study	Country/ Study Period	Total Number of iCCA/cHCC-CCA with LDLT	iCCC	cHCC-CCA	Neoadjuvant Therapy	Underlying Disease	Overall Survival 1, 3 and 5 Years	Recurrence-Free Survival 1, 3 and 5 Years
1	Hara T. et al.	Japan 2001–2005	19	13	6	TACE	Cirrhosis no further description	79%/63%/46%	79%/45%/45%
2	Serra V. et al.	Italy 2000–2015	1	1	0	TACE, RFA	PSC		
3	Itoh S. et al.	Japan 1999–2014	8	0	8	n.a.	-	87.5%/72.9%/72.9%	85.7%/85.7%/85.7%
4	Park Y.-H. et al.	South Korea 1999–2009	14	0	15	TACE	14× HBV 1× NTLC	66.7%/60%/60%	60%/53.3%/53.3%
5	Fukuda A. et al.	Japan 2012	1	1	0	no	Biliary atresia/ Kasai Op	died	died
6	Song S. et al.	South Korea 1995–2012	7	0	7	n.a.	HBV	n.a./n.a./50%	n.a./n.a./37.5%
7	Nart D. et al.	Turkey 2012	2	2	0	n.a.	1× HBV 1× HCV	n.a.	n.a.
8	Chan A. et al.	China 2002–2003	2	0	3	n.a.	1× HBV 1× HCV	n.a.	100%/100%/33.3%
9	Vilchez V. et al.	USA 1994–2013	6	0	6	n.a.	-	82%/47%/40%	n.a./93%/n.a.
10	Togashi J. et al.	Japan 1996–2015	3	1	2	yes	-	80%/n.a./78%	5%/6%/6%
11	Chang C. et al.	Taiwan 2006–2014	11	0	11	yes	-	90%/61.7%/n.a.	80%/46.7%/n.a.
12	Jonas S. et al.	Germany 1999–2004	2	2	0	-	Liver fibrosis	n.a.	n.a.
13	Sotiropoulos G.C. et al.	Germany	1	1	0	-	Recurrence after resection		
14	Takatsuki M et al.	Japan 1997	1	1	0	-	Caroli	alive	no recurrence
15	Hafeeq Bhatti AB et al.	Pakistan 2012–2019	16	9	7	yes		63.6%/n.a./63.6%	n.a./46.7%/n.a.
16	Schwenk L et al.	Germany 2010–2022	1	6	4	yes	-	87.5%/75%./75%	85.7%/85.7%/85.7%

## Data Availability

The data can be shared up on request.

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
