# Peer review of "Liver Transplantation for Incidental Cholangiocarcinoma or Combined Hepatocellular Carcinoma/Cholangiocarcinoma—Own Experiences and Review of the Literature"

_cancers, 2023, doi:10.3390/cancers15143609_

Round 1

Reviewer 1 Report

 Discussion about LT for ICC/PHCC is the hottest topic for transplant surgeons.  Your reports can be one of milestones of LT for ICC/PHCC. Please revise along with my comment.

1) I think surgical difficulty is easier compared to usual LC cases because ICC/PHCC patients usually do not have coagulation abnormality and severe collateral vessel developments. From thes backgrounds, please add perioperative data.

2) Table 4 No.4 is not Japan.

3) Please add the Kaplan-Meier curve of patients without LT (staging

Author Response

Discussion about LT for ICC/PHCC is the hottest topic for transplant surgeons.  Your reports can be one of milestones of LT for ICC/PHCC. Please revise along with my comment.

1) I think surgical difficulty is easier compared to usual LC cases because ICC/PHCC patients usually do not have coagulation abnormality and severe collateral vessel developments. From thes backgrounds, please add perioperative data.

We added perioperative data, as requested by the reviewer.

2) Table 4 No.4 is not Japan.

We thank the reviewer for this comment. It was changed.

3) Please add the Kaplan-Meier curve of patients without LT (staging

Since all our findings were incidental carcinoma, we are not able to add a survival analysis for non-transplanted patients.

Reviewer 2 Report

The authors presented an interesting review and retrospective analysis study about liver transplantation for incidental cholangiocarcinoma (iCCA) or combined hepatocellular carcinoma/cholangiocarcinoma (cHCC-CCA) with valuable data that are worth publishing.  They included eight patients with incidental iCCA and four with cHCC-CCA. They accurately selected statical analysis tests for examining their hypothesis, given the small sample size of included patients. They found that iCCA patients could benefit from liver transplantation. These findings may be valuable input for better practice and health outcomes for iCCA patients, although studies with large sample sizes would be required to further support their hypothesis.

The paper requires minor proofreading. 

Author Response

The authors presented an interesting review and retrospective analysis study about liver transplantation for incidental cholangiocarcinoma (iCCA) or combined hepatocellular carcinoma/cholangiocarcinoma (cHCC-CCA) with valuable data that are worth publishing.  They included eight patients with incidental iCCA and four with cHCC-CCA. They accurately selected statical analysis tests for examining their hypothesis, given the small sample size of included patients. They found that iCCA patients could benefit from liver transplantation. These findings may be valuable input for better practice and health outcomes for iCCA patients, although studies with large sample sizes would be required to further support their hypothesis.

We thank the reviewer for this comment.

Round 2

Reviewer 1 Report

The authors have revised along with the reviewer's suggestions.